# OpenReview forum: "On the Overlooked Structure of Stochastic Gradients"
_NeurIPS.cc/2023/Conference — NeurIPS 2023 poster_

### Official Review · Reviewer_AgKN · 2023-06-24

**Soundness:** 3 good
**Presentation:** 3 good
**Contribution:** 2 fair
**Rating:** 6
**Confidence:** 3

**Summary:**

This paper studies two overlooked structures of stochastic gradients in deep learning. Specifically, the first part of the study conducts formal statistical tests on the distribution of stochastic gradients across parameters and iterations, and empirically showed that dimension-wise gradients exhibit power-law heavy tails, while iteration-wise gradient noise often has Gaussian-like light tails. Then, the second part of the study discovers that the covariance of stochastic gradients has the power law spectra in deep learning, which is overlooked by previous studies. The power-law gradient covariance may help understand the success of stochastic optimization for deep learning.

**Strengths:**

This paper conducts comphrehensive deep learning experiments to reveal the structures of stochastic gradients and the corresponding covariance matrix.

The experiments show that dimension-wise gradients exhibit power-law heavy tails, while iteration-wise gradient noise often has Gaussian-like light tails. This addresses an existing conflict arguments on the distribution of stochastic gradients in the existing literature.

These new empirical observations may help understand the success of stochastic optimization for deep learning and may inspire new theoretical developments.

**Weaknesses:**

N/A

**Questions:**

1. It seems that the main observations of this paper also hold for random initialized deep models. While this shows that these structures may be a fundamental and common structure of deep networks, does it imply that these structures may not be induced by the SGD training?

2. How is the gradient matrix G computed for random initialized models and pre-trained models? Do you train these models for some iterations and compute the G?

---

> ### Author Rebuttal · Authors · 2023-08-08
>
> We sincerely thank Reviewer AgKN for kind support and comments.
>
> We also duly address your questions as follows.
>
> Q1: It seems that the main observations of this paper also hold for random initialized deep models. While this shows that these structures may be a fundamental and common structure of deep networks, does it imply that these structures may not be induced by the SGD training?
>
> A1: Thanks for the question. We think these structures are induced by both stochastic training and network architectures. Because, without SGD training, the full-batch gradient has no such structure; without proper network architectures, the power-law gradient structure may be broken. Interestingly and similarly, the success of deep learning can also attribute to both stochastic training and network architectures.
>
> Q2: How is the gradient matrix G computed for random initialized models and pre-trained models? Do you train these models for some iterations and compute the G?
>
> A2: Thanks for the question. We present the details in Appendix A.1, Lines 12-15. We freeze the model weights at some given point and only compute the gradients for $T$ iterations. The gradients over $T$ iterations can form the $n\times T$ matrix $G$.
>
> The reviewer definitely recognized how the contributions of our work can significantly inspire the community. We gratefully hope the reviewer can insist on your opinions.

---

> > ### Comment · Reviewer_AgKN · 2023-08-21
> > **comment**
> >
> > Thanks for the response. I will keep the score.

---

### Official Review · Reviewer_tTSf · 2023-06-30

**Soundness:** 3 good
**Presentation:** 3 good
**Contribution:** 3 good
**Rating:** 6
**Confidence:** 3

**Summary:**

This paper focuses on investigating the stochastic gradient noise (SGN), defined as the difference between the full-batch gradient and the stochastic gradient. The authors conduct formal statistical tests to examine the distribution of SGN across both parameters and iterations. Their analysis reveals that dimension-wise SGN often demonstrates power-law heavy tails, while iteration-wise SGN resulting from mini-batch training generally lacks such heavy tails. Additionally, the authors discover power-law structures in the covariance spectra of stochastic gradients. By means of formal statistical tests and covariance spectra analysis, the authors contribute to a deeper comprehension of the behavior and properties of stochastic gradients within the context of deep neural networks.

**Strengths:**

1. Distinguishing between two distinct types of SGNs is important, providing a clearer understanding of distribution regarding to SGN.

2. The power-law behavior observed in the covariance spectrum, along with the theoretical implications and insights it offers regarding the low-dimensional learning space of deep neural networks (DNNs), is very interesting.



**Weaknesses:**

1. The experiments conducted in this study primarily focus on LeNet (a CNN with approximately 60,000 parameters) and FCN models, especially when analyzing the covariance spectrum due to limited computational source. It is important to note that the observations made on these relatively simple DNN models may not necessarily apply to more complex architectures such as ResNet or Transformer.

2. I also believe that the authors have missed an important line of research that examines the eigenvalue spectrum of the Hessian and the gradient covariance, accompanied by empirical findings. I find that their empirical results are closely connected to the discoveries presented in this paper, particularly regarding the power-law nature of the covariance spectrum.

[1] Sagun, Levent, et al. "Empirical analysis of the hessian of over-parametrized neural networks." arXiv preprint arXiv:1706.04454 (2017).
[2] Papyan, Vardan. "The full spectrum of deepnet hessians at scale: Dynamics with sgd training and sample size." arXiv preprint arXiv:1811.07062 (2018).
[3] Papyan, Vardan. "Measurements of three-level hierarchical structure in the outliers in the spectrum of deepnet hessians." arXiv preprint arXiv:1901.08244 (2019).
[4] Li, Xinyan, et al. "Hessian based analysis of sgd for deep nets: Dynamics and generalization." Proceedings of the 2020 SIAM International Conference on Data Mining.



**Questions:**

1. Is the plot made based on multiple runs or just a single run?
2. Is the plot shown at initialization, in the middle of the training, or at the end of training (close to a local minima)?The authors have discussed the potential limitation of this work.




**Limitations:**

The authors have discussed the potential limitation of this work.

---

> ### Author Rebuttal · Authors · 2023-08-08
>
> We sincerely appreciate Reviewer tTSf for kind support and helpful comments.
>
> We duly address your concerns as follows.
>
> Q1: The experiments conducted in this study primarily focus on LeNet (a CNN with approximately 60,000 parameters) and FCN models, especially when analyzing the covariance spectrum due to limited computational source. It is important to note that the observations made on these relatively simple DNN models may not necessarily apply to more complex architectures such as ResNet or Transformer.
>
> A1: Thanks for the suggestion. Due to memory and computation capacity, we indeed mainly conducted experiments on CNNs and FCNs, while the gradient structure of ResNet18 is presented in Figure 3. We sincerely acknowledge your suggestion. We will try to scale our experiments to larger models.
>
>
> Q2: I also believe that the authors have missed an important line of research that examines the eigenvalue spectrum of the Hessian and the gradient covariance, accompanied by empirical findings. I find that their empirical results are closely connected to the discoveries presented in this paper, particularly regarding the power-law nature of the covariance spectrum.
>
> A2: Thanks for providing these references (mainly on the Hessian structures). The Hessian structure and the gradient structure are indeed closely related. We will discuss the references in the paper.
>
> Q3: Is the plot made based on multiple runs or just a single run?
>
> A3: We have run the experiments for multiple times and obtain consistent conclusions on the gradient structure. The visualized plots are mainly based a single run. Because, as we visualize the magnitude and the magnitude rank in the plots, averaging over multiple runs may break the meaning of the eigenvalues’ magnitude rank.
>
> Q4: Is the plot shown at initialization, in the middle of the training, or at the end of training (close to a local minima)?
>
> A4: We analyzed randomly initialized models and pretrained models (terminated at the end of training and close to a minimum). The gradient structure holds very similarly. Please see Figures 3-4 and Tables 1-2. Moreover, we believe it will hold similarly to models in the middle training phase, as this middle phase simply interpolates between the two studied training phase.

---

> > ### Comment · Reviewer_tTSf · 2023-08-14
> >
> > I would like to thank the authors for their rebuttal. After carefully considering the rebuttal and the other reviews, I have chosen to keep my score.

---

### Official Review · Reviewer_8wue · 2023-07-02

**Soundness:** 3 good
**Presentation:** 3 good
**Contribution:** 3 good
**Rating:** 5
**Confidence:** 4

**Summary:**

This paper looks at the structure of stochastic gradient noise (SGN).  They show that the noise for a single parameter, across minibatches/iterations is light-tailed, while looking across parameters, the noise is heavy-tailed.  Further, they show that the spectrum across parameters is heavy-tailed.

**Strengths:**

Interesting collection of empirical results.

**Weaknesses:**

* In many settings, the proposed power-law structure is violated, with N-1 large and similarly-sized eigenvalues, where N is the number of image classes.  There is no theory on this point.
* In several settings, there are large deviations between the Hessian and FIM.  There is no theory on this point.
* The theory in "Robust and low-dimensional learning subspace." is problematic.  It starts from the observation of power-law gradients.  However, by far the most important theory, in the context of this paper, would explain the emerge of power-law gradients.
* The fact that the power-law gradients seem to emerge only at sufficient width strongly suggests that theory is possible.

This is bizarre.
> Is it possible that the number of outliers depends on the number of model outputs (logits) rather than the number of data classes? In Figure 12, we eliminate the possibility by training a LeNet with 100 logits on CIFAR-10, denoted by CIFAR-10⋆. The number of outliers   will be constant even if we increase the model logits.
I'm not at all sure that it is meaningful to have more logits than classes, and as such, the conclusion is not supportable.

Ultimately the AC decision will come down to whether such a collection of empirical results, with no theory describing how or why they might emerge should be published.

**Questions:**

N/A

---

> ### Author Rebuttal · Authors · 2023-08-08
>
> We sincerely appreciate Reviewer 8wue for hard work and recognizing main contributions of our work.
>
> We notice that Reviewer 8wue agrees on the contribution of our finding and the associated empirical and statistical evidences, while the main concern about lacking of theory is posted.
>
> We frankly admit that we currently have no formal theory for explaining why the power-law stochastic gradient (SG) structure exist. The SG structure is beyond the understanding of existing theories and has been overlooked until our work. We will leave formal theories for explaining the SG structure as future work.
>
> However, in this work, we have tried hard to (1) develop theoretical implication of the novel SG structure for explaining and understanding deep learning behaviors and (2) how various settings of DNNs can empirically affect the SG structure.
>
> For example, we formulate theoretical analysis for explaining and analyzing robust and low-dimensional learning subspace of DNNs via the SG structure and Davis-Kahan Theorem. We empirically revealed how the SG structure behaves differently under extensive experimental settings. We think our empirical and theoretical contributions are much more than simply reporting existence of the novel SG structure.
>
> Most importantly, we and even all five reviewers consistently believe that the overlooked structure of stochastic gradients is indeed novel and important for our community.
>
> We also duly address your main concerns as follows.
>
> Q1: In many settings, the proposed power-law structure is violated, with N-1 large and similarly-sized eigenvalues, where N is the number of image classes. There is no theory on this point.
>
> A1: We agree that no improper theory is available yet in our work or related work. Our contribution on this finding (Section 5.5) is that the number of large outliers not only closely depends on the number of data classes. It is the number of data classes rather than the number of the final layer’s logits that decide the number of large outliers. We further show that BatchNorm can make the large outliers less significant. This may also explain why BatchNorm can stabilize training and work with a large learning rate. This finding is also valuable.
>
> Q2: In several settings, there are large deviations between the Hessian and FIM. There is no theory on this point.
>
> A2: We respectfully argue that large deviations between the Hessian and FIM can be explained by falsely assuming that the model is near a critical point.
>
> Our empirical results actually suggest that the approximation of Hessian and FIM is very weak along the "sharp" directions corresponding to large eigenvalues, while previous works suggest the approximation is robust along the ``flat” directions corresponding to small eigenvalues. The assumption that the model is near a critical point is only mild for those flat directions.
>
> Q3: The theory in "Robust and low-dimensional learning subspace." is problematic. It starts from the observation of power-law gradients. However, by far the most important theory, in the context of this paper, would explain the emerge of power-law gradients.
>
> A3: It indeed does not explain emergence of the SG structure. We respectfully argue that, in this section, we only aim at explaining and analyzing robust and low-dimensional learning subspace via the SG structure. We think the relations of the SG structure to other behaviors of DNNs are important.
>
> Q4: The fact that the power-law gradients seem to emerge only at sufficient width strongly suggests that theory is possible.
>
> A4: We sincerely thank for the constructive suggestion. We will explore its theoretical mechanism and implication given the observed relations to width and nonlinearity.
>
> Q5: About Figure 12 in Section 5.5. I'm not at all sure that it is meaningful to have more logits than classes, and as such, the conclusion is not supportable.
>
> A5: We respectfully argue that the experiment of Figure 12 is meaningful, because the backpropagated gradients always start from the final layer’ logits. While Figure 11 suggests that $C-1$ large outliers exist in the spectra, where $C$ is the number of data classes, this phenomenon may also depend on the final network layer’s logits. Thus, it is meaning full to have this further ablation study, where the backpropagated gradients also start from all 100 logits but the number of data classes is still 10.

---

> > ### Author Response · Authors · 2023-08-17
> > **Discussion**
> >
> > We hope our responses could address at least some of your concerns.
> >
> > We will appreciate it if the reviewer may consider our rebuttal content as well as other reviewers' opinions during discussion.
> >
> > If there are any further questions, we are glad to continue the discussion.

---

### Official Review · Reviewer_jrig · 2023-07-06

**Soundness:** 2 fair
**Presentation:** 3 good
**Contribution:** 4 excellent
**Rating:** 7
**Confidence:** 4

**Summary:**

This paper studies some statistical properties of stochastic gradients, with a particular focus on their covariance.
The main finding is that the stochastic gradients have an approximately Gaussian distribution whose covariance spectrum follows a power law, which is robust to several changes in the setting.

EDIT: I have read the author's rebuttal, which partially addressed my concerns. However, some points still require clarification or a more precise analysis.

**Strengths:**

This paper studies a very important and central problem.
Besides the finding that the gradient covariance has a power-law structure, it makes several interesting observations, notably the fact that it persists with random labels and that it occurs in a linear network as soon as one adds batch normalization.

**Weaknesses:**

One main weakness is that I find it hard to understand what exactly is being plotted in Figure 1. The distinction between rows and columns of the gradient history matrix is clear. However, I don't understand how the collection of rows/columns is being mapped to a single scatter plot. The authors should add a precise description of the considered quantity, perhaps in the form of an equation.

The authors claim that the stochastic gradient noise across iterations is Gaussian. I disagree with this claim, as it seems to me that stochastic gradient noise should be defined for a batch size of 1, as the difference between the gradient computed on a *single* data point and the full-batch gradient. The Gaussianity here seems to only come from averaging with a sufficiently large batch size, as shown in Figure 2. So the Gaussianity is not an inherent statistical property of stochastic gradients, but only an outcome of the central limit theorem. I think this distinction is important and should be clarified in the text. As a side remark, the Gaussianity "rate" cannot be larger than 95% as the significance level (rate of false rejections) is 0.05.

The power-law fits for ResNet are erroneous: they should be corrected or removed (see limitations). One issue is that this figure considers only 200 timesteps. It implies that eigenvalues beyond the top, say, 100, are not well estimated, which makes the presence of a power law difficult to assess. The authors acknowledge the memory costs of storing the large $n \times n$ covariance matrix. However, they seem to have missed that one can compute the eigenvalues of the covariance from the singular values of the much smaller $n \times T$ gradient history matrix. The large number of parameters $n$ could also be reduced by considering gradients with respect to a single layer, which can be expected to also show a power law structure. I am not necessarily requesting that this be added in this paper, which already has enough results to be intesting, but the limitation of the ResNet results should be acknowledged by the authors.

Finally, I do not see the point of the study of the eigengaps. The experiments amount to showing that the derivative of a power law is another power law, with an index one lower, which is elementary. I am also not convinced by the "stability" analysis. The authors define stable as being roughly invariant during training, but they do not study the evolution of the Hessian as the parameters remain frozen in their experiments. Besides, the fact that the eigenvectors associated with small and/or close eigenvalues are unstable to perturbations is a well-known general fact, which has nothing to do with stochastic gradient descent or power laws.

Let me suggest some relevant references. The work of Papyan [1,2] studies the spectrum of the Hessian and shows the presence of outliers created by the class structure of the cross-entropy loss. These works are relevant to the discussion in paragraph 5. of Section 5. The work of Sagun et al. [3, 4] is an early investigation into the eigenvalues of the Hessian, noting in particular that it has many small eigenvalues at the end of training (the power law was not observed due to the linear scaling of the axes in the scree plots).

Typos:
- Figure 5: spectrums -> spectra
- line 226: eliminate "which"
- The caption of Figure 13 wrongly states that Adam preserves the power law

In general, the quality of the English could be improved.

[1] Papyan, Vardan. "Measurements of Three-Level Hierarchical Structure in the Outliers in the Spectrum of Deepnet Hessians." International Conference on Machine Learning. PMLR, 2019.

[2] Papyan, Vardan. "Traces of class/cross-class structure pervade deep learning spectra." The Journal of Machine Learning Research 21.1 (2020): 10197-10260.

[3] Levent Sagun, Leon Bottou, and Yann LeCun. Eigenvalues of the Hessian in deep learning: Singularity and beyond. arXiv preprint arXiv:1611.07476, 2016.

[4] Levent Sagun, Utku Evci, V Ugur Guney, Yann Dauphin, and Leon Bottou. Empirical analysis of the Hessian of over-parametrized neural networks. arXiv preprint arXiv:1706.04454, 2017.

**Questions:**

What is being plotted in Figure 1? On what are the $\chi^2$-KS tests exactly computed?

What are the authors trying to show with the study of the eigengaps?

**Limitations:**

One limitation of the work is that it is restricted to relatively small-scale experiments, mostly limited to small networks and/or datasets. See weaknesses for a few suggestions for scaling up the experiments.

---

> ### Author Rebuttal · Authors · 2023-08-08
>
> We sincerely appreciate Reviewer jrig’s kind support and constructive comments.
>
> We duly address your concerns as follows.
>
> Q1: One main weakness is that I find it hard to understand what exactly is being plotted in Figure 1. The distinction between rows and columns of the gradient history matrix is clear. However, I don't understand how the collection of rows/columns is being mapped to a single scatter plot. Or what are the $\chi^{2}$-KS Test exactly computed.
>
> A1: We apologize for the confusion. A scatter point in Figure 1 indicates an element of a dimension-wise or iteration-wise gradient vector. The vertical axis indicates the magnitude of a gradient element, and the horizontal axis indicates the rank order of the gradient element. As visualizing the whole matrix G is not clear and informative, we visualize and statistically analyze only one dimension-wise gradient vector and one iteration-wise gradient vector in Figure 1. Other figures have no such confusion, because we only visualize and statistically analyze the eigenvalues. We will make this point more clear in the revision.
>
> Q2: The authors claim that the stochastic gradient noise across iterations is Gaussian. I disagree with this claim, as it seems to me that stochastic gradient noise should be defined for a batch size of 1, as the difference between the gradient computed on a single data point and the full-batch gradient. The Gaussianity here seems to only come from averaging with a sufficiently large batch size, as shown in Figure 2. So the Gaussianity is not an inherent statistical property of stochastic gradients, but only an outcome of the central limit theorem. I think this distinction is important and should be clarified in the text.
>
> A2: We agree that the Gaussianity of stochastic gradient noise (SGN) mainly come from Central Limit Theorem. However, we also note that, a large body of related works usually discuss SGN with arbitrarily large or small batch sizes, as long as the optimizer is SGD not GD. Thus, it is not false to claim that SGN across iterations can be approximately Gaussian (due to Central Limit Theorem). We mentioned this point in Lines 140-145. We will follow your suggestion and make the distinction more clear.
>
> Q3: The power-law studies of ResNet are limited. One issue is that this figure considers only 200 timesteps. Authors seem to have missed that one can compute the eigenvalues of the covariance from the singular values of the much smaller gradient history matrix. The large number of parameters could also be reduced by considering gradients with respect to a single layer, which can be expected to also show a power law structure. I am not necessarily requesting that this be added in this paper, which already has enough results to be interesting, but the limitation of the ResNet results should be acknowledged by the authors.
>
> A3: We sincerely appreciate the constructive comment. We will acknowledge the limitation of the current ResNet18 experiment in the paper. We agree that it is possible to calculate a more precise eigenspectrum corresponding to a larger gradient history matrix G for ResNet18 with some computational tricks. We will also try to scale our analysis up to larger network architectures in the revision.
>
> Q4: I do not see the point of the study of the eigengaps. I am also not convinced by the "stability" analysis. The authors define stable as being roughly invariant during training, but they do not study the evolution of the Hessian as the parameters remain frozen in their experiments. Besides, the fact that the eigenvectors associated with small and/or close eigenvalues are unstable to perturbations is a well-known general fact, which has nothing to do with stochastic gradient descent or power laws.
>
> A4: We respectfully argue that the study of the eigengaps can help us understand robust and low-dimensional learning subspace of DNNs reported in previous studies, because learning subspace directly relates to the gradient structure. Our discussion aims at the gradient structure rather than the Hessian structure.
>
> We understand that the existence of small eigengaps is common. However, it is not well known until our work that the eigengaps of stochastic gradients exhibit power laws which indicate the existence of certain top eigengaps. Moreover, the top eigengaps ``happen” to correspond to top eigenvalues, where learning dynamics mainly happens. These evidences together can support our analysis of robust and low-dimensional learning space.
>
> Without the power-law gradient structure, there are other possibilities: (1) large eigengaps do not exist, so no learning subspace is stable; (2) large eigengaps exist but do not correspond to large eigenvalues, so learning dynamics happens in other unstable subspace instead of the much more stable subspace.
>
> Q5: Some relevant references.
>
> A5: Thanks for providing these references on the Hessian structures. The Hessian structure and the gradient structure are indeed closely related. We will discuss the references in the paper.
>
> Q6: Typos.
>
> A6: Thanks again for pointing out the typos. We will correct them.
>
> We gratefully thank Reviewer jrig again for recognizing the significant contribution of our work to the community and the very helpful suggestions.

---

> > ### Comment · Reviewer_jrig · 2023-08-10
> >
> > I thank the authors for their detailed answers.
> >
> > If I understand A1, Figure 1 plots the ordered entries of a single row/column of the gradient history matrix. Observing a power law here is then a priori different from having a power law on the eigenvalues of the covariance matrix. The basis is indeed different, from the canonical basis of the parameters in Figure 1 to the covariance eigenbasis in the other figures. How do the authors interpret these two findings together?
> >
> > With the modifications announced by the authors (some clarifications and acknowledgments of limitations), I recommend that the paper be accepted. It makes several fundamental observations that are of important interest to the NeurIPS community.
> >
> > I am still not convinced about the eigengaps study, which seems to me as a restatement of standard random matrix theory results for power-law spectra. However, it is not the core of the paper, which remains of significant value even without this section.

---

### Official Review · Reviewer_HTRt · 2023-07-06

**Soundness:** 3 good
**Presentation:** 3 good
**Contribution:** 3 good
**Rating:** 6
**Confidence:** 4

**Summary:**

This paper makes two contributions. First, it conducts certain formal statistical tests on the distribution of stochastic gradients and gradient noise across both the parameters and iterations. Second, it  further discover that the covariance spectra of stochastic gradients have the power-law structures overlooked by previous studies and present its theoretical implications for training of DNNs.

**Strengths:**

This paper conduct the formal  statistical tests to show that dimension-wise gradients usually exhibit power-law heavy tails, while iteration-wise gradients and stochastic gradient noise caused by minibatch training usually do not exhibit power-law heavy tails. It further discover that the covariance spectra of stochastic gradients have the power-law structures and present its theoretical implications for training of DNNs. These results are quite significant for pratical applications.

**Weaknesses:**

The experimental results are not so sufficient. The models utilized in the experiments are LeNet (LeCun et al., 1998), Fully Connected Networks (FCN), and ResNet18 (He et al., 2016). However, it is better to conduct the xperiments on other network structures, such as LSTM and Transformer.


**Questions:**

A theoretically explanation as to why the power-law covariance generally exists in deep learning should be provided. Furthermore, when is this statement true and when is it false?


**Limitations:**

The two theoretical results are important and significant. The authors should invesigate how these results can applied in the deep learning architectures.

---

> ### Author Rebuttal · Authors · 2023-08-08
>
> We sincerely appreciate Reviewer HTRt’s kind support and constructive comments.
>
> We duly respond to your comments as follows.
>
> Q1: The experimental results are not so sufficient. The models utilized in the experiments are LeNet (LeCun et al., 1998), Fully Connected Networks (FCN), and ResNet18 (He et al., 2016). However, it is better to conduct the experiments on other network structures, such as LSTM and Transformer.
>
> A1: Thanks for the suggestion. We will add more experiments in the revision. We also note that the overlooked structure of stochastic gradients is general and interesting as it emerges in even very simple neural networks.
>
> Q2: A theoretical explanation as to why the power-law covariance generally exists in deep learning should be provided. Furthermore, when is this statement true and when is it false?
>
> A2: We frankly admit that we currently have no formal theory for explaining why the power-law stochastic gradient (SG) structure exist. The SG structure is beyond the understanding of existing theories and has been overlooked until our work. We will leave formal theoretical analysis for explaining the SG structure as future work.
>
> However, in this work, we have tried hard to develop theoretical implication of the novel SG structure for explaining and understanding deep learning behaviors. For example, we formulate theoretical analysis for explaining and analyzing robust and low-dimensional learning subspace of DNNs via the SG structure and Davis-Kahan Theorem.
>
> Q3: The two theoretical results are important and significant. The authors should investigate how these results can applied in the deep learning architectures.
>
> A3: Thanks again for the constructive suggestion and recognizing the value of our theoretical results. We agree that your suggestion will be a promising exploration direction. For example, the robust and low-dimensional learning subspace may inspire novel low-rank efficient training method, while this is beyond the main scope of this work. We will actively explore related topics in future.

---

### Decision · Program_Chairs · 2023-09-21

**Decision:**

Accept (poster)

**Comment:**

This paper studies stochastic gradient descent in trianing deep neural networks. with a particular focus on their covariance. The authors perform statistical tests on the distribution of stochastic gradients, and then demonstrate that the covariance spectrum of stochastic gradients follows a power law, which is robust to several changes in the setting. The results are solid and all reviewers have recommended acceptance.